# Advancing the application of pXRF for animal samples

**Kate J. Brandis**[1]*, **Roxane J. Francis**[1], **Kyle J. A. Zawada**[2], **Chris D. Hasselerharm**[2], **Daniel Ramp**[2]

**1** Centre for Ecosystem Science, School of Biological, Earth and Environmental Sciences, University of New South Wales, Sydney, Australia, **2** Centre for Compassionate Conservation, University of Technology Sydney, Broadway, Australia

* Kate.Brandis@unsw.edu.au

**Data Availability Statement:** All relevant data are within the manuscript and its Supporting Information files.

**Funding:** The author(s) received no specific funding for this work.

## Abstract

Portable x-ray fluorescent (pXRF) technology provides significant opportunities for rapid, non-destructive data collection in a range of fields of study. However, there are sources of variation and sample assumptions that may influence the data obtained, particularly in animal samples. We used representative species for four taxa (fish, mammals, birds, reptiles) to test the precision of replicate scans, and the impact of sample thickness, sample state, scan location and scan time on data obtained from a pXRF. We detected some significant differences in concentration data due to sample state, scanning time and scanning location for all taxa. Infinite thickness assumptions were met for fish, reptile and mammal representatives at all body locations. Infinite thickness was not met for feathers. Scan time results found in most cases the 40, 60 and 80 second beam scan times were equivalent but significantly different to 20 second beam scan times. Concentration data across replicate scans were highly correlated. The opportunities for the use of pXRF in biological studies are wide-ranging. These findings highlight the considerations required when scanning biological samples to ensure the required data are suitably collected and standardised while reducing radiation exposure to live animals.

## Introduction

Elemental and isotopic signature analyses are routine in the biological and environmental sciences, being used as indicators of diet [1,2], ecotoxicology [3], pollution [4], soil sciences [5] and animal movement studies [6,7]. Current analytical techniques include stable isotope analysis (SIA), inductively coupled plasma mass spectrometry (ICP-S), atomic absorption spectroscopy (AAS), and neutron activation (NAA) among others [8]. However, many of these techniques are destructive, time consuming, and expensive, precluding their use in the field, or on live organisms. Development of rapid, *in situ*, inexpensive and non-destructive sampling techniques would greatly expand the utility of elemental signatures in biological and environmental sciences.

X-ray fluorescence (XRF) is a well-established methodology for measuring the elemental composition of samples [9]. It is a quantitative, non-destructive technique that measures the

**Competing interests:** The authors have declared that no competing interests exist.

abundance of an element based on the characteristic emission of secondary x-rays following excitation by a primary x-ray beam [10]. Portable x-ray fluorescence (pXRF) instruments provide opportunities for field-based data collection. Developed initially for use in the field of geology [11], they are also used in a diverse range of fields including archaeology [12–14], art history [15,16], forensics [17,18] and food sciences [19,20] among others [21–23].

Data obtained from XRF is also used for a range of applied biological questions including the detection of in vivo disease for assessment in humans [24,25], toxicology in humans [26–28] and wildlife [29,30], and geographic provenance for wildlife [7,31]. However, it's use in animal studies, has been limited relative to the opportunities it offers.

The portability, non-destructive and rapid multi-variable data collection offers significant opportunities in the field of animal sciences. However, due to its initial development for use in geology there are assumptions about sample characteristics that are not necessarily met by biological, and other non-geological samples. This has potential implications for the way in which samples should be processed and scanned, how raw data is processed by pXRF on-board algorithms, and the resulting data provided to the user.

These assumptions include the thickness and density of the sample, referred to as 'infinite thickness', i.e., the sample is of sufficient thickness and density that all x-rays are contained within the sample and do not pass through it [32,33]. It has been recognized that data acquired from samples that are thin, and do not meet infinite thickness need to be corrected [33]. The thickness required to meet this assumption varies with the density of the material being analysed [34], which is challenging to accurately determine in living samples.

Moisture content is another variable that can impact on the results returned by pXRF [35,36]. Water absorbs the characteristic x-rays from the elements and causes the primary radiation from the excitation sources to scatter. This results in a decrease in the intensity of characteristic x-rays and an increase in the intensity of scattered x-rays in a fluorescence spectrum [37]. Higher moisture content in soils has been shown to impact data resulting in lower elemental concentrations [35]. Moisture content of biological samples is highly varied and dependant on sample state, and in living samples moisture content is very difficult to determine.

The length of time a sample is exposed to x-rays can impact on the data obtained. Scanning times may affect the ability to detect low-concentration or low atomic weight elements, where longer scan times improve the signal-to-noise ratio [38]. However, in a biological context, there may be a trade-off between detecting elements, and exposing a live organism to excessive radiation. As such, determining whether elemental signatures significantly change in biological samples as a function of scanning time is a key consideration to optimise data quality and organism welfare.

Further, the location of the scan on the sample can also influence results. The partitioning of elements in biological samples may occur due to tissue type, metabolic processes [39], age, and sex [40]. Studies by Nganvongpanit et al. (2016) found different elemental concentrations for different bones within the same individual while Buddhachat et al., (2016) found differences between tissue types with keratinous tissues (hair, nail, skin), being different to bone.

The use of portable X-ray fluorescence (pXRF) as a non-destructive technique for elemental analysis of biological samples offers significant opportunities in the field of biological and environmental sciences. However, when planning a sampling design for biological samples, there are a number of potential sources of variation that need to be accounted for to ensure data are comparable and research questions are answerable. This paper aimed to explore the application of pXRF for elemental analysis of biological samples and to test the major sources of variation across four representative taxa, namely infinite thickness, sample state (dried/thawed), scanning time, and scanning location on the sample. The findings of this study provide insights for future studies and will contribute to the development of standardised protocols for pXRF analysis of biological samples.

## Materials and methods

### Sampling design

We chose species representing four taxa commonly studied in biological research: reptiles, mammals, birds and fish. Representative species for each taxa were chosen based upon accessibility to sufficient specimens, with each species represented by ten samples.

Reptiles were represented by the shingleback lizard (*Tiliqua rugosa*) and were sourced as whole or gutted carcasses from Taronga Conservation Society, Macquarie University and Murdoch University. Mammals were represented by the European rabbit (*Oryctolagus cuniculus*) and sourced from a pet food supplier. Fish were represented by Eastern school whiting (*Sillago flindersi*) sourced from the Sydney Fish Markets, Australia. Birds were represented by Australian maned duck (*Chenonetta jubata*), and moulted flight feathers were collected from wetland sites. All taxa except the duck feathers were stored frozen prior to scanning. Each of the taxa were used to test each of the sources of variation.

### pXRF data collection

An Olympus Vanta M-Series portable x-ray fluorescence (pXRF) instrument (4-watt X-ray tube with tungsten (W) anode, 8-50keV, silicon drift detector), with three beam energies (10, 40 and 50keV) was used to scan all specimens. Where size permitted, specimens were scanned within the Olympus workstation [41]. Samples were scanned using the GeoChem3 method which uses a fundamental parameters algorithm [42] that automatically corrects for inter-element effects [41]. The instrument was factory calibrated by Olympus and calibration checks were performed throughout the sampling process using the onboard calibration option. The Vanta pXRF provides two output data types; raw beam spectra containing information across 2048 keV bands, and elemental concentrations as a percentage for 42 elements calculated via Olympus' on-board algorithms.

Reptile, mammal, and fish specimens were scanned in both thawed and dried states, feathers were scanned in their natural state. All specimens were scanned at 20, 40, 60, and 80s/beam with and without silica backing at multiple locations (Fig 1) and states (Fig 2) with 2 replicate scans. The silica dioxide block backing consisted of 30 mm diameter x 25 mm solid cylinder. Prior to scanning, specimens were defrosted at room temperature for ~12 hours and scanned thawed. Specimens were then oven dried (60˚C for 72 hours) and re-scanned.

### pXRF data analyses

Data from the Vanta were exported as JSON files and processed in R [43] to convert them to compiled csv files for analysis. Files were first read into R using the jsonlite [44] package and relevant data extracted into a dataframe using the tidyverse [45] package. Concentration and raw beam spectra data were extracted from the JSON data and processed using the xrftools [46] package. Spectra processing included associating known characteristic fluorescence energies for elements to the correct spectra energy range, baseline measurement and reduction, and applying a gaussian smoothing filter.

To ensure that elements analysed were present in the samples we checked for diagnostic peaks in the beam spectra data using 80s scans for each species group (thawed state). We excluded any elements that did not match their Kα1 and Kß2 peaks [47]. This approach allowed us to remove all non-biologically relevant elements. Eighty second scan data, the longest scan time in this study, were chosen as providing the greatest possibility of detecting the element if present. Elements below the limits of detection were not included in analyses.

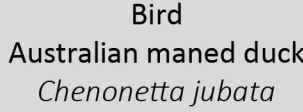
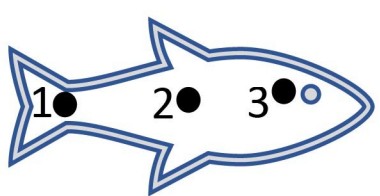
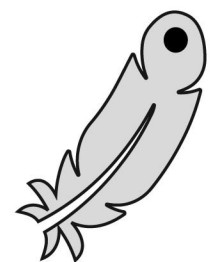
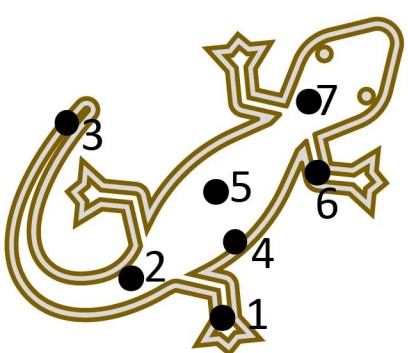
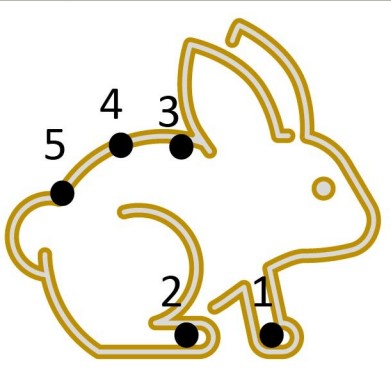

**Fig 1. Scanning locations (black dots) on each specimen of representative taxa.**

To test whether infinite thickness was being met we inspected the beam spectra data to analyse the dispersion of the characteristic lines of the X-ray tube, in this case, the L line for tungsten (W). Where characteristics lines were the same for samples scanned with and without backing, we determined that infinite thickness had been met.

Sample state, scanning time, and scan-location were all tested using a fitted a median quantile regression for each scan location for each species using the concentration data (20, 40, 60 and 80s/beam), and were split into an overall difference test across all elements and a per-element analysis to identify specific element differences. Median quantile regression was chosen as it is more robust than standard least-squares linear regression to skew and kurtosis in the residuals [48]. In cases with multiple response categories such as scanning location or scanning time, pair-wise models were run to, for example, compare leg to tail, leg to head, head to tail, etc. Per-element analyses followed the median quantile approach described above. Overall differences were tested using multi-response generalised linear models [49], or manyglm's. Manyglm's allow a model to be specified with multiple response variables (in this case the concentration of each element in a sample) and to calculate an overall effect of an explanatory variable such as tail vs head scanning location. This approach examines all elemental differences simultaneously, and provides overall statistics, including an overall p.value for the effect

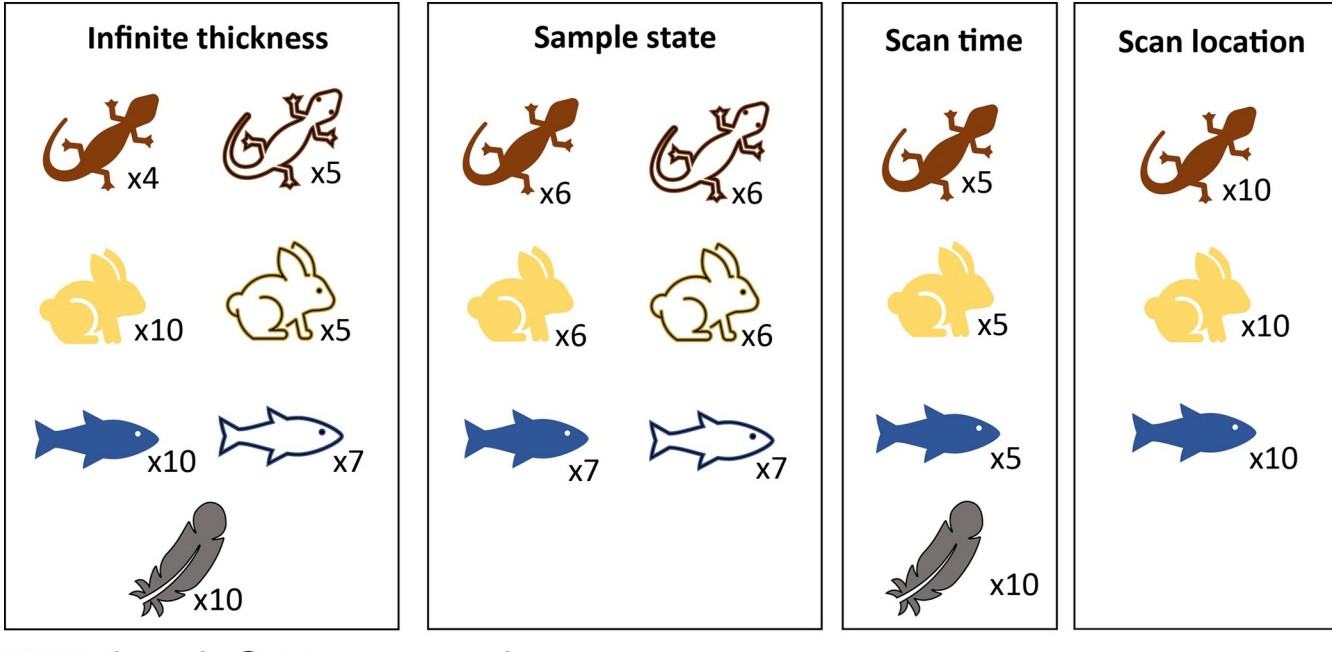

**Fig 2. Summary of the number of individuals specimens per variation test and condition; natural state, thawed or dried.**

of the explanatory variable. Unfortunately, no median quantile version of a manyglm is available at present, meaning that these models rely on the assumptions of linear regression. Despite this, they are still likely more robust than typical distance-based approaches [49].

We also tested for correlation between replicate scans to determine the precision of the pXRF when working with biological samples, using Spearman's correlation coefficient.

## Results

We excluded 22 elements from our analyses (Y, Nb, Ag, In, Sb, Te, La, Ta, W, Pt, Au, Pb, Bi, Si, V, Cr, Co, Ce, Ne, Pd, U and Th) due to absences in their Kɑ1 and/or Kß2 peaks.

Infinite thickness assumptions were met for thawed and dried states for fish, rabbits and lizards using concentration data (Fig 3). Infinite thickness was not met for feathers using either concentration and beam spectra data (Fig 3).

Overall, we detected a mix of significant ($p < 0.05$) and non-significant ($p > 0.05$) differences in concentration data due to sample state (Fig 3 and S1 Table), scan time (Fig 3 and S2 Table) and scan location for all taxa (Fig 3 and S3 Table).

All three taxa (reptiles, fish, and mammals) tested for differences between dried and thawed sample states found significant differences in the concentration data overall, with multiple per-element differences identified (Fig 3 and S1 Table).

Scanning time had a significant effect on overall detected elemental concentrations for all taxa when comparing the 20 second beam time to the 40, 60 and 80 second beam times. (Fig 3). All three taxa tested for differences between scan locations revealed some significant differences in overall elemental concentration data amongst differing scan locations (Fig 3).

Overall device precision was very high, with a Spearman's correlation coefficient of 0.999 when comparing replicate scans). There was variation in the correlation of replicate scans across elements, with heavier elements less correlated than lighter elements (S4 Table).

| Infinite thickness | Sample state | Scan time | Scan location |
|---|---|---|---|
| Infinite thickness met at all 7 body locations.<br><br>Infinite thickness met at all 7 body locations. | Significant differences between thawed and dried; Ca, Cu, LE, P, Rb, Sr, Zn. (Supplementary Dataset 1) | Scan times ≥40s equivalent. Significant differences between Ba, Cu and Zr at 20s/beam when compared to 40, 60 & 80 s/beam. (Supplementary Dataset 2).<br><br>Mg, Mn significantly different (p<0.05) 20-40s, Mg, Mn, Zr, Se 20-60s, Mn, Cu, Rb 20-80s. | Significant difference between all scan locations excluding positions 5-2, 5-4, 1-6, 5-2, 7-2, 5-7 (Fig 1; Supplementary Dataset 3).<br><br>Significant difference between all scan locations excluding positions 4-2, 5-2, 1-6, 4-7, 5-7, 5-4, 7-2 (Fig 1; Supplementary Dataset 3). |
| Infinite thickness met at all 5 body locations.<br><br>Infinite thickness met at all 5 body locations. | Significant differences between thawed and dried; Al, Ca, Cu, Fe, K, LE, P, Rb, S, Sr, Zn, Zr. | Mn, Al significantly different (p<0.05) for 20-80s, Al 20-40s, Mn, Al 20-60s, Mn 40-60s, 40-80s.<br><br>Mn, Ti significantly different (p<0.05) for 20-80s, Mn 20-60s, Mn 40-60s, Mn, Ti 40-80s. | Significant differences between all scan locations (Supplementary Dataset 3).<br><br>Significant differences between all scan locations except between positions 1 & 2 (Supplementary Dataset 3). |
| Infinite thickness met at all 3 body locations.<br><br>Infinite thickness met at all 3 body locations. | Significant differences between thawed and dried; Al, As, Ca, Cu, Fe, K, LE, Mg, P, Rb, S, Sr, Zn, Zr | Mo, Se significantly different (p<0.05) for 20-80s, Mo 20-60s, Mn 40-60s, S for 40-80 scan time comparisons<br><br>Mn, Mo, Se significantly different (p<0.05) for 20-80s, Mn, Se, Mo 20-60s, Mn 40-60s, 40-80s | Significant differences between all scan locations (Supplementary Dataset 3).<br><br>Significant differences between all scan locations (Supplementary Dataset 3). |
| Infinite thickness not met | Not tested | No significant differences | Not tested |

🔴🟡🔵 thawed    ⚪ dried    ⚫ natural state

**Fig 3. Summary of results for each test using concentration data.**

## Discussion

In this study we investigated the application of pXRF for sampling biological samples, focusing on the effects of sample thickness, sample state, scanning time, and scan location on elemental concentration data. Our results found that sample state, and scan location resulted in the most significant differences between element concentrations. Infinite thickness assumptions were met for most tests regardless of sample state, excluding feather samples. Scan time results found in most cases the 40, 60 and 80 second beam times were equivalent.

These findings are consistent with previous studies that reported the impacts of sample moisture [35,50], and sample thickness [34,35] on pXRF data collected from soil samples. Extended scan times were not found to result in significant data gains, which is supported by findings by Williams, Taylor [51], however is contrary to findings by Zhang, Specht [38], which found extended (5 minute) scan times reduced variability in results when scanning bone through soft tissue. Extending scan times on live specimens increases radiation exposure which can result in health impacts [52]. Estevam and Appoloni [24] calculated that a 50 second scan using an x-ray source of 13 and 17 keV resulted in an exposure of 3 mSV.

Based on the results of this study and considering animal care and ethics we would recommend when scanning live specimens that scan times are kept to a minimum while still

ensuring adequate data is obtained, our study suggests that 40 s/beam is sufficient to achieve this for most elements (Fig 3). We would also recommend scanning a part of the body that is of sufficient thickness to meet infinite thickness assumptions, but away from key organs to limit the impact of radiation exposure, and to scan with a standardised and consistent backing behind the specimen whenever possible. For the Olympus Vanta this is the provided silicon dioxide block but any other inert, elementally simple material should suffice. This study found that on thawed specimens (akin to live organisms with regards to body thickness and moisture) the majority of scan locations met infinite thickness requirements (Fig 3). However, noting that different scan locations on the specimen may results in different results (Fig 3). This is an important consideration if wanting standardised data between specimens. To ensure consistency the same location should always be scanned. Lastly, noting that we found significant differences in data due the state (thawed or dried) of the sample.

There are some limitations of this study that should be considered when translating our results. Firstly, we only tested a limited number of biological samples from a small set of taxa. Our results may not be generalizable to other types of biological samples but are intended as a guide to the impact different sample characteristics, and sampling techniques may have on results. Secondly, we used a single pXRF instrument with a specific configuration and settings. Different instruments or settings may produce different results [53], however the broad effects demonstrated here should hold as the sources of variation are extrinsic to the instrument. Finally, we did not evaluate the long-term stability or repeatability of pXRF measurements for biological samples [12]. Future research should address these issues by testing more diverse biological samples and elements, comparing different pXRF instruments and settings, and assessing the quality control and assurance procedures for pXRF analysis.

This study demonstrates that pXRF can be a useful tool for biological research if sample characteristics and sample design are cognisant of the assumptions of the pXRF device, and differences within and between samples as highlighted in this study. The advantages of pXRF over conventional analytical techniques include its portability, non-destructiveness, speed, and capability to measure many elements simultaneously. These features make pXRF an attractive option for biological studies that require in situ or large-scale analysis of elemental signatures in various types of samples, such as feathers, hair, bones, plants, soils, etc. However, our study also highlights the considerations required when sampling biological samples using pXRF and the potential impact on results obtained, which include where to scan, how long to scan and how to prepare the sample.

## Supporting information

**S1 Table. Sample state test results.** Significant (p<0.05) results of sample state testing. (CSV)

**S2 Table. Scan time test results.** Significant (P<0.05) results of scan time testing. (CSV)

**S3 Table. Scan location test results.** Significant (p<0.05) results of location test results. (CSV)

**S4 Table. Spearman rank results.** Spearman correlation co-efficients for device precision. (CSV)

## Acknowledgments

B. Jackson, Murdoch University, WA. P. Meagher, Taronga Conservation Society, D. Harasti, NSW Department of Planning and Environment, The Reptile Lab, Macquarie University.

## Author Contributions

**Conceptualization:** Kate J. Brandis, Daniel Ramp.

**Data curation:** Roxane J. Francis, Kyle J. A. Zawada.

**Formal analysis:** Kate J. Brandis, Roxane J. Francis, Kyle J. A. Zawada, Daniel Ramp.

**Investigation:** Kate J. Brandis, Roxane J. Francis, Chris D. Hasselerharm.

**Methodology:** Kate J. Brandis, Roxane J. Francis.

**Supervision:** Kate J. Brandis, Daniel Ramp.

**Writing – original draft:** Kate J. Brandis.

**Writing – review & editing:** Kate J. Brandis, Roxane J. Francis, Kyle J. A. Zawada, Daniel Ramp.

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
