## [Decision Letter · Decision Letter 0]

20 Feb 2024

PONE-D-24-01800ADVANCING THE APPLICATION OF pXRF FOR BIOLOGICAL SAMPLESPLOS ONE

Dear Dr. Brandis,

Thank you for submitting your manuscript to PLOS ONE. After careful consideration, we feel that it has merit but does not fully meet PLOS ONE’s publication criteria as it currently stands. Therefore, we invite you to submit a revised version of the manuscript that addresses the points raised during the review process.

We look forward to receiving your revised manuscript.

Kind regards,

Dawid Surmik

Academic Editor

PLOS ONE

Journal Requirements:

2. We notice that your supplementary figures and tables (Appendix 1 and 2) are included in the manuscript file. Please remove them and upload them with the file type 'Supporting Information'. Please ensure that each Supporting Information file has a legend listed in the manuscript after the references list.

Additional Editor Comments:

Dear Authors,

I received reviews and although one of them rejected the article, in my opinion the work is worth correcting and clarifying and clarifying any doubtful issues.

Please make the necessary changes in accordance with the reviewers' recommendations and respond to their comments point by point.

Sincerely yours,

Dawid Surmik

Reviewers' comments:

Reviewer's Responses to Questions

**Comments to the Author**

1. Is the manuscript technically sound, and do the data support the conclusions?

Reviewer #1: Yes

Reviewer #2: No

2. Has the statistical analysis been performed appropriately and rigorously? 

Reviewer #1: Yes

Reviewer #2: No

3. Have the authors made all data underlying the findings in their manuscript fully available?

Reviewer #1: Yes

Reviewer #2: Yes

4. Is the manuscript presented in an intelligible fashion and written in standard English?

Reviewer #1: Yes

Reviewer #2: Yes

5. Review Comments to the Author

Reviewer #1: The article titled "ADVANCING THE APPLICATION OF pXRF FOR BIOLOGICAL SAMPLES" is undeniably an important work that addresses the application of portable X-ray fluorescence (pXRF) technology for analyzing biological samples, focusing on animals. It is clear from the review that this study is necessary and holds substantial value for publication due to its contributions to the field. However, several areas require attention and improvement to enhance the clarity, accuracy, and overall impact of the work.

Firstly, the title suggests a broader review (biology samples) rather than an experimental study limited to animal tissues. It would be beneficial to refine the title to accurately reflect the scope of the samples analyzed or to adjust the content to match the broad implication of the current title. This would ensure readers have a clear expectation of the article's content and scope.

The introduction lacks a comprehensive overview of the current literature on the application of pXRF in both biological and other types of samples, missing an opportunity to place the study within the context of existing research. Citing key studies, such mentioned below, would provide a solid background for readers:

https://doi.org/10.1016/j.trac.2023.117355

https://doi.org/10.1016/j.trac.2023.117165

https://doi.org/10.1016/j.jhazmat.2023.132167

The methodological details appear to need more depth regarding the pXRF technique's application.

Please remember that portable X-ray fluorescence is not just a point and shoot method (see great review: https://doi.org/10.1016/j.envint.2019.105250)

Questions arise from the description provided, such as:

- the standards used for calibration,

- measurement modes,

- duration of measurements,

- how the data, including fluorescence spectra and detection limits (LOD, LOQ), were handled?

Supplementary materials are not sufficient, readers want to know step by step methodology workflow and all related data (full excel with level, time duration, mode of measurement and etc).

Addressing these questions would significantly strengthen the credibility and reproducibility of the study.

Additionally, the presentation of Latin names with spell-check underlines visible in Figure 1 suggests a need for a more professional approach to figure preparation. Utilizing dedicated graphic design software could enhance the visual quality of the publication.

While this study is fundamentally significant and intriguing, a major revision is required to address these concerns thoroughly. Enhancing the title's specificity, expanding the literature review, detailing the methodology more comprehensively, and improving figure quality would greatly benefit the manuscript's contribution to the field and its suitability for publication.

Reviewer #2: The paper describes an analysis of different alternatives for the application of pXRF in biological tissues, aiming to show the simplicity of the technique and the wide range of potential applications. This point has been well established in the work, clearly showing that there are uses of pXRF in biology that have not yet been explored. Effectively, pXRF is very simple to use, but without careful analysis of the measurements, it is limited to a qualitative, usually preliminary study of more detailed laboratory studies. This work has some weaknesses, specifically in the data analysis.

On the one hand, the method used to identify infinite thickness samples needs to be improved as it is not sufficiently accurate. Effectively, if the silicon signal is successfully detected, then the sample is thin for silicon and for elements with atomic weight greater than silicon, as X-ray attenuation in the sample decreases as Z increases. However, if the silicon signal is not detected, it is difficult to conclude that the sample is infinite for two reasons: the XRF spectrometer's detection limit is low, and X-ray attenuation in the sample decreases as Z increases. In the first case, a high silicon detection limit leads to the conclusion that the sample is infinite when it is not actually the case. In the second case, because X-ray attenuation decreases drastically with Z, a thickness that may be considered infinite for silicon may not be so for heavier elements like iron, for example.

In summary, the methodology proposed by the authors effectively identifies samples of very thin thickness but does not allow for the precise identification of samples of infinite thickness. Therefore, it can erroneously endorse the elemental quantification of non-infinite thickness samples with the model of infinite thickness, leading to an underestimation of elemental concentrations. This error is serious because it may conclude that there are variations in the sample's composition when in reality, it's just variations in its thickness. One way to obtain more information about the sample thickness is to analyze the dispersion of the characteristic lines of the X-ray tube, in this case, the W L lines (See for example A. A. Markowicz and R. E. Van Grieken, Chapter 8. Quantification in XRF Analysis of Intermediate-Thickness Samples, in Handbook of x-ray spectrometry, Marcel Dekker, New York, 2002, 2nd edn.)

In other hand, when there are many overlapped XRF lines, many commercial software infer the possible presence of elements based on fractions of theoretical lines, which must be carefully reviewed. In this work has been reported elements that are rarely found in biological tissues, such as U, Th, Pr, La, Pb, Sb, Sn, Bi. I recommend including some representative XRF spectra of the samples to allow observation of which elements are directly detected and only using these elements in subsequent statistical analysis.

6. PLOS authors have the option to publish the peer review history of their article (what does this mean?). If published, this will include your full peer review and any attached files.

Reviewer #1: No

Reviewer #2: No

---

## [Author Response · Author response to Decision Letter 0]

6 Jun 2024

Please see uploaded detailed response

Comments to the Author

1. Is the manuscript technically sound, and do the data support the conclusions?

Please see our detailed responses to reviewers’ comments below, we feel we have addressed reviewers 2’s valid concerns and the manuscript now meets this criterion.

2. Has the statistical analysis been performed appropriately and rigorously?

Please see our detailed responses to reviewers’ comments below, we feel we have addressed reviewers 2’s valid concerns and the manuscript now meets this criterion.

5. Review Comments to the Author

We have changed the title of the manuscript as suggested. It is now “ADVANCING THE APPLICATION OF pXRF FOR ANIMAL SAMPLES

As per the reviewers comments we have expanded the introduction from lines 58 – 64 to include additional key references and latest publications in this area. We have included the references noted by the reviewer and some additional, and recent studies that help to demonstrate range of applications of pXRF in both biological and other non-biological fields. 

We concur with the reviewer, the pXRF is not just a ‘point and shoot’ instrument, hence our manuscript identifying the type of issues that need to be considered when sampling animal tissues. We feel that our methods as described are adequate and repeatable. We have added some text to the methods section (line 146 onwards) regarding the analysis of diagnostic peaks and the determination of infinite thickness based on Reviewer 2’s comments.

 The Olympus Vanta is a self calibrating device. Calibration checks where done regularly throughout the testing process. The following text has been added at Line 126 “The instrument was factory calibrated by Olympus and calibration checks were performed throughout the sampling process using the onboard calibration option.” 

 Line 124 states the GeoChem3 mode was used.

 Line 132 states the duration of measurements

Any samples returning a measure below the LOD were removed from analyses. This has been noted in the text at Line 151.

We understand the reviewers’ comments, however we would argue that the step by step methodology is presented in the text. We had added some additional methods on data handling at lines 147-153.

All related data extends to over 495,000 data points for concentration data and 12,500,000 data points for the raw beam spectra data which we have noted is available on request from the authors, but we don’t feel if useful as a supplementary data set. We have provided key data in the supplementary that supports our results. 

These errors have been corrected and visual quality improved. All figures have been recreated using Adobe Illustrator updated and replaced.

We thank the reviewer for their time and comments, we hope we have adequately addressed them.

Reviewer #2: 

We are very grateful for the reviewer bringing these issues to our attention, it has significantly improved our manuscript and we have addressed both issues raised. We took the reviewers advice and reanalysed our data. To reassess if infinite thickness was being accurately detected we inspected the beam spectra data to analyse the dispersion of the characteristic lines of the X-ray tube, in this case, the L line for tungsten (W). We have revised as results as required based on the new analyses. We have amended our methods to remove reference to using silica to determine infinite thickness and now use reviewer 2’s suggested method (lines 153-156). We have also amended our results as necessary.

With regards to Reviewer 2’s comment on non-biologically relevant elements. Again, we reanalysed our data and checked for diagnostic peaks in the beam spectra data using 80s scans for each species group. We excluded any elements that did not match their Kɑ1 and Kß2 peaks. This approach allowed us to remove all non-biologically relevant elements. We have amended our methods to describe this approach (lines 146-150). We have also amended our results as necessary. 

---

## [Decision Letter · Decision Letter 1]

15 Aug 2024

ADVANCING THE APPLICATION OF pXRF FOR ANIMAL SAMPLES

PONE-D-24-01800R1

Dear Dr. Brandis,

We’re pleased to inform you that your manuscript has been judged scientifically suitable for publication and will be formally accepted for publication once it meets all outstanding technical requirements.

Kind regards,

Dawid Surmik, PhD

Academic Editor

PLOS ONE

Additional Editor Comments (optional):

Reviewers' comments:

Reviewer's Responses to Questions

**Comments to the Author**

1. If the authors have adequately addressed your comments raised in a previous round of review and you feel that this manuscript is now acceptable for publication, you may indicate that here to bypass the “Comments to the Author” section, enter your conflict of interest statement in the “Confidential to Editor” section, and submit your "Accept" recommendation.

Reviewer #2: All comments have been addressed

Reviewer #3: All comments have been addressed

2. Is the manuscript technically sound, and do the data support the conclusions?

Reviewer #2: Yes

Reviewer #3: Yes

3. Has the statistical analysis been performed appropriately and rigorously? 

Reviewer #2: Yes

Reviewer #3: Yes

4. Have the authors made all data underlying the findings in their manuscript fully available?

Reviewer #2: Yes

Reviewer #3: Yes

5. Is the manuscript presented in an intelligible fashion and written in standard English?

Reviewer #2: Yes

Reviewer #3: Yes

6. Review Comments to the Author

Reviewer #2: The authors properly addressed my concerns regarding the manuscript and thus improved the paper. I recommend to accept the new version of the manuscript.

Reviewer #3: (No Response)

7. PLOS authors have the option to publish the peer review history of their article (what does this mean?). If published, this will include your full peer review and any attached files.

Reviewer #2: No

Reviewer #3: No

---

## [Editor Report · Acceptance letter]

22 Aug 2024

PONE-D-24-01800R1 

PLOS ONE

Dear Dr. Brandis, 

I'm pleased to inform you that your manuscript has been deemed suitable for publication in PLOS ONE. Congratulations! Your manuscript is now being handed over to our production team.

Kind regards, 

on behalf of

Dr. Dawid Surmik 

Academic Editor

PLOS ONE